# Selection and Characterization of Phosphate-Solubilizing Fungi and Their Effects on Coffee Plantations

**DOI:** 10.3390/plants12193395

**Published:** 2023-09-26

**Authors:** Rosa María Arias, Gabriela Heredia Abarca, Yamel del Carmen Perea Rojas, Yadeneyro de la Cruz Elizondo, Karla Yosselín García Guzman

**Affiliations:** 1Instituto de Ecología A. C., Carretera Antigua a Coatepec, No. 351. Col. El Haya, Xalapa 91070, Veracruz, Mexico; rosa.arias@inecol.mx; 2Centro de Investigación en Micologia Aplicada, Universidad Veracruzana Médicos No. 5, U.H. del Bosque, Xalapa-Enríquez 91017, Veracruz, Mexico; yperearojas@gmail.com; 3Facultad de Biología, Universidad Veracruzana, Campus Xalapa, Circuito Gonzalo Aguirre Beltrán s/n. CP Zona Universitaria, Xalapa 91090, Mexico; ydelacruz@uv.mx; 4Instituto Tecnológico Superior de Xalapa, Reserva Territorial SN, Col. Santa Bárbara, Xalapa 91096, Mexico; karlayosselingarciaguzman@gmail.com

**Keywords:** filamentous fungi, *Coffea arabica* var. Costa Rica, phosphorous, *Penicillium brevicompactum*, coffee bean production

## Abstract

The use of phosphate-solubilizing fungi in coffee cultivation is an alternative to the use of traditional fertilizers. The objective of this study was to analyze the mechanisms involved in the phosphorus solubilization of fungal strains and to evaluate the effect of a phosphate-solubilizing strain on coffee plants. For this, phosphorus-solubilizing fungal strains were selected for evaluation of their solubilization potential and phosphatase activity. Coffee plants were inoculated in the field with a phosphate-solubilizing strain, and the soil and foliar soluble phosphorus contents, as well as coffee bean yield, were quantified. Of the 151 strains analyzed, *Sagenomella diversispora*, *Penicillium waksmanii*, and *Penicillium brevicompactum* showed the highest solubilization. *Aspergillus niger* and *P. waksmanii* presented the highest soluble phosphorus values; however, *P. brevicompactum* showed the highest phosphatase activity. The *P. brevicompactum* strain inoculated on the coffee plants did not favor the foliar phosphorus content but increased the soil soluble phosphorus content in two of the coffee plantations. The plants inoculated with the phosphate-solubilizing strain showed an increase in coffee bean weight on all plantations, although this increase was only significant in two of the three selected coffee plantations.

## 1. Introduction

Cultivated coffee is considered to be the world’s leading agricultural commodity, with a market generating more than USD 90 billion annually. Around 8% of the world’s population (approximately 500 million people) is involved in the coffee market, from sowing to final consumption. Coffee is produced in 14 states of Mexico, of which Chiapas, Veracruz, Puebla, and Oaxaca account for 90% of the production. In 2019, Chiapas contributed 40.9% of the national production, followed by Veracruz with 24.2%, Puebla with 16.0%, and Guerrero with 9.4% [1]. The state of Veracruz is the second most important coffee-producing region in Mexico. The coffee is produced in 842 communities of 82 municipalities. Sixty percent of the coffee is grown above 750 MSL in elevation. The regions with the highest production are Coatepec, Cordoba, Huatusco, Misantla, and Atzalan. Currently, 21,089 producers in the central region of Veracruz produce this crop over a total area of 58,712 ha, representing 7.3% of the area dedicated to coffee growing nationally. The rural development district of Coatepec cultivates 28,873 hectares in 21 municipalities. Jilotepec belongs to this district and has a planted area of 1776 hectares [2].

Coffee growing is considered to be a fundamental strategic activity because it allows integration of productive chains, as well as generating foreign currency and employment, and it is a means of subsistence for many small producers and indigenous groups [3,4]. In addition, it is of great ecological importance since more than 90% of the area cultivated with this crop is under shade, which contributes to biodiversity conservation and provides vital environmental services to society [5,6]. In Mexico, the soils where coffee is produced are generally of volcanic origin and are characterized by an acidic pH of 4.5–5.2, as well as a low availability of essential macronutrients such as phosphorus (P) [7]. This is mainly because this element is associated with other ionic elements such as calcium, iron, and aluminum, present in forms that cannot be assimilated by the plants [8]. The amount of available P depends on modification of the dynamic equilibrium that maintains the dissolution of insoluble inorganic compounds and on decomposition of organic matter [9]. Phosphorus is important in coffee cultivation since it plays key roles in many plant processes such as energy metabolism, the synthesis of nucleic acids and membranes, photosynthesis, respiration, nitrogen fixation, and enzyme regulation [10]. During the early stages of coffee plant development, this element is responsible for vigorous plant growth, participates in the formation of effective root systems, and acts as a promoter of flowering and fruit development. During the reproductive stage, phosphorus is essential for the formation, growth, and multiplication mechanisms of the flower organs. Phosphorus deficiency in the soil can be observed in coffee plants through the yellowing of leaves and a lack of fruit ripening. In severe cases, the leaves near the ripening fruit fall off completely. To remedy a lack of P, coffee producers apply large amounts of phosphate fertilizers [1], although 75–90% of the phosphate added precipitates through the formation of metal cation complexes [11]. In addition, excessive use of fertilizers leads to eutrophication, water toxicity, groundwater contamination, air pollution, soil and ecosystem degradation, biological imbalances, and reduced biodiversity, so it is necessary to seek another option to release P from inorganic and organic pools of the total soil P. One alternative is the application of bioinoculants, which are preparations of microorganisms for inoculation with the aim of partially or completely replacing inorganic fertilization.

In this context, phosphate-solubilizing fungi (PSF) are of great importance since they are a functional group of microorganisms that play a fundamental role in the P cycle. Thanks to the activity of these fungi, plants can take advantage of the large reserves of insoluble phosphorus fixed to soil minerals [12]. Some fungi and bacteria can solubilize P from unavailable forms in the rhizosphere. The mechanism of mineral phosphate solubilization by these strains is associated with the release of low molecular weight organic acids [13], which, through their hydroxyl and carboxyl groups, act to chelate the cations bound to phosphate, thereby converting it into soluble forms. Phosphate-solubilizing fungi can produce extracellular enzymes, i.e., a group of phosphatase enzymes that can mineralize organic P into inorganic P such that P becomes available for the plants. There are several soil phosphatases, the most common of which are phosphomonoesterases, phosphodiesterases, and phytases. Phosphomonoesterases act on phosphate monoesters and, according to their optimum pH, are categorized into acid and alkaline phosphomonoesterases. These microorganisms have a high potential for promoting plant growth by increasing soil fertility [14]. In agricultural soils, the use of microbial inoculants (biofertilizers) that present phosphate-solubilizing activities is considered to be an environmentally friendly alternative to further applications of chemical-based P fertilizers [11]. The coffee rhizosphere is associated with many beneficial organisms, including growth-promoting microorganisms that may contribute to fulfilling the nutritional requirements of the plant. The objectives of this study were to evaluate PSF, to obtain further information about the mechanisms involved in the solubilization of P, and to evaluate the effect of inoculating PSF strains on coffee plants under field conditions. 

## 2. Results

### 2.1. Selection of Strains with Phosphate-Solubilizing Capacity

Of the 151 soil fungal strains tested, 120 strains (79.5%) formed solubilization halos (Figure 1). Considering the established scales of solubilization, 55 strains (45.8%) were found to be in level I, 39 strains (32.5%) in level II, and 26 strains (21.6%) in level III. Level I (halos 1–4 mm in thickness) was represented by 24 genera including *Trichoderma*, *Oidiodendron*, *Penicillium*, *Chaetomium*, and *Humicola*; level II (halos of 5–9 mm in thickness) was represented by 12 genera including *Penicillium*, *Aspergillus*, *Fusarium*, *Cylindrocarpon*, and *Talaromyces*; level III (halos of 9–12 mm in thickness) was represented by 13 genera, most prominently *Penicillium*, *Aspergillus*, *Fusarium*, and *Eupenicillium* (Figure 1).

According to the maximum values obtained for the SI, and based on the scale used by Silva Filho and Vidor [15], six of the strains evaluated had low solubilizing activity. The species in this group are *Aspergillus sclerotiorum*, *Aspergillus sydowii*, *Aspergillus* sp., *Humicola* sp., *Merimbla* sp., and *Penicillium glabrum*. A further 15 species presented medium solubilization: *Scopulariopsis brevicaulis*, *Aspergillus candidus*, *A. niger*, *Fusarium* sp. 25, *Fusarium* sp. 3Y, *Eupenicillium euglaucum*, *Eupenicillium ludwigii*, *Talaromyces flavus* var. flavus, *Cladosporium cladosporioides*, *Acremonium roseolum*, *Epicoccum nigrum*, *Penicillium arenicola*, *Penicillium* sp., *Penicillium olsonii*, *Penicillium verruculosum*, and *Penicillium miczynskii* (Figure 2).

All species presented a clear halo zone around their colonies, and *Sagenomella diversispora*, *Trichocladium asperum*, *Penicillium waksmanii*, and *Penicillium brevicompactum* each presented a maximum solubilization index. The highest phosphate solubilization rate was observed for the strains *Sagenomella diversispora* (3.51), *P. waksmanii* (3.88), and *P. brevicompactum* (3.61) (*p* < 0.05). However, for *Aspergillus niger,* the phosphate solubilization index was 2.6 and the clear zone became visible on the first day, while the clear zone of the other fungal isolates became visible on the third day.

### 2.2. Phosphate Solubilization and pH in Sundara

Significant differences were detected in the soluble phosphorus values among the extracts of the PSF strains evaluated (F = 50.31, *p* = 0.0005). The available phosphorus content in the extracts of the treatments corresponding to the PSF strains was significantly higher than that of the control treatment (no PSF) (*p* < 0.05). The available phosphorus concentration varied over the period of evaluation; however, a higher content was observed on Day 21 for all strains. The soluble phosphorus values of the extracts of the *A. niger* and *P. waksmanii* strains (98.22 mg/L and 95.77 mg/L, respectively) were significantly (*p* < 0.05) higher than that of *P. brevicompactum* (74.63 mg/L) (Figure 3).

The pH of the extracts where the PSF strains (A. niger, *P. brevicompactum*, and *P. waksmanii*) were inoculated decreased significantly (*p* < 0.05) with respect to the pH of the control treatment. The pH of the extracts of the PSF strains was variable between the days of incubation. The pH decreased from 7 to 1.9 for *A. niger*, from 7 to 2.9 for *P. brevicompactum*, and from pH 7 to 3.0 for *P. waksmanii* (Figure 4). 

### 2.3. Acid Phosphatases

Regarding phosphatase enzyme activity, significant differences in this enzyme were observed among the strains evaluated (F = 63.86, *p* = 0.001). Production of acid phosphatase in the *P. brevicompactum* strain was significantly higher than that detected for the rest of the strains tested (*A. niger* and *P. waksmanii*) and the control treatment (no PSF).

The highest acid phosphatase production was observed in the extract of the *P. brevicompactum* strain on Days 15 and 18 (189.75 and 176.94 UAE mg/protein, respectively). In the *A. niger* strain, this was detected on Days 15 and 21 (59.44 and 69.59 UAE mg/protein, respectively), while in *P. waksmanii*, the highest acid phosphatase activity was found on Day 21 (38.81 UAE mg/protein) (Figure 5).

Three bands were detected in the results of the acid phosphatase isoenzyme patterns: one monomorphic (1) and two polymorphic (2 and 3). Three different isoenzyme patterns were also observed: The first showed two bands (2 and 3) for the *A. niger* strain. The second pattern exhibited two bands (1 and 3) for the *P. waksmanii* strain. The third pattern exhibited one band (3) for the *P. brevicompactum* strain (Figure 6).

### 2.4. Coffee Plant Foliar Phosphorus

At the San Isidro coffee plantation, the initial foliar phosphorus content in the plants was 1112.96 mg/kg. Sixty days after establishing the experiment, the foliar phosphorus content in the plants inoculated with the PSF strain (*P. brevicompactum*) was 1528.12 mg/kg, while in the non-inoculated plants, this value was 1695.31 mg/kg. At 120 days, there was a decrease in the foliar phosphorus content of both the PSF-inoculated and non-inoculated plants (933.27 and 1366.18 mg/kg, respectively). At 180 days, there was an increase in the leaf phosphorus content in both the inoculated (1456.95 mg/kg) and non-inoculated (1557.29 mg/kg) plants. In all cases, these differences were not significant (*p* > 0.05) (Figure 7A).

At the “Los Bambus” coffee plantation, for the two plots, the initial foliar phosphorus contents in the plants were, on average, 1113 and 1118 mg/kg, respectively. None of the samplings (at 60, 120, and 180 days) revealed significant differences between the inoculated and non-inoculated plants (*p* > 0.05). At 60 days, the average leaf phosphorus content in the plants inoculated with the PSF strain (*P. brevicompactum*) was 1353 mg/kg, while in the non-inoculated plants, this value was 1406 mg/kg. At 120 days, there was a slight decrease in the leaf phosphorus content of the inoculated (1215 mg/kg) and non-inoculated (1311 mg/kg) *P. brevicompactum* plants. At 180 days, a significant increase in the leaf phosphorus content was observed in both the plants inoculated with the *P. brevicompactum* strain (2164 mg/kg) and the non-inoculated plants (2315 mg/kg) (Figure 7B).

At the “La Barranca” coffee plantation, for the two plots, the average initial foliar phosphorus contents in the plants were 998 and 992 mg/kg. No significant differences were detected in the leaf phosphorus content between the plants from the two plots (inoculated and non-inoculated) at 60, 120, and 180 days (*p* > 0.05).

At 60 days, the leaf phosphorus content of the plants inoculated with the PSF strain (*P. brevicompactum*) averaged 1287 mg/kg, while that of the non-inoculated plants averaged 1342 mg/kg. At 120 days, there was a slight decrease in the foliar phosphorus content of both the inoculated (981 mg/kg) and non-inoculated (1007 mg/kg) *P. brevicompactum* plants. At 180 days, a significant increase in the leaf phosphorus content was observed in both plots of plants inoculated with the *P. brevicompactum* strain (1824 mg/kg) and the non-inoculated plants (2115 mg/kg) (Figure 7C).

### 2.5. Soil Phosphorus Available to the Coffee Plants

At the “San Isidro” coffee plantation, the initial soil soluble phosphorus content was 4.6 mg/kg. Sixty days after inoculation with the phosphate-solubilizing fungus *P. brevicompactum*, an increased soil soluble phosphorus content of 5.64 mg/kg was observed, while in the soil of the non-inoculated plants, this value was 4.74 mg/kg. However, these increases were not significant (F = 0.017, *p* = 0.89). At 120 days, a decrease in the soluble phosphorus content was detected in the soil of both the inoculated and non-inoculated (3.82 and 2.30 mg/kg, respectively) *P. brevicompactum* plants, being significantly higher in the inoculated plants. By 180 days, the available phosphorus had increased significantly, and the content of this element was higher in the plants inoculated with the PSF (4.98 mg/kg) (Figure 8A).

At the “Los Bambus” coffee plantation, the soil presented 2.93 mg/kg of initial soluble phosphorus. Sixty days after inoculation with the PSF *P. brevicompactum*, an increase in the soluble phosphorus content to 5.08 mg/kg was observed, while the content in the soil of the non-inoculated plants increased to 3.27 mg/kg. At 120 days, a decrease in the soluble phosphorus content was detected in the soil of both the inoculated and non-inoculated plants (2.65 and 1.79 mg/kg, respectively). At 180 days, the soluble phosphorus content in the soil of the inoculated plants was 3.32 mg/kg, while that of the non-inoculated plants was 2.31 mg/kg. In all three samplings, the measured soluble phosphorus content was significantly higher in the *P. brevicompactum*-inoculated plants than in the non-inoculated plants (*p* < 0.05) (Figure 8B).

At the “La Barranca” coffee plantation, the available phosphorus content in the soil was 0.93 mg/kg. In the first sampling at 60 days after inoculation with the phosphate-solubilizing fungus *P. brevicompactum*, an increase in the soluble phosphorus content in the soil to 1.22 mg/kg was observed, while the content in the soil of the non-inoculated plants increased to 1.14 mg/kg. At 120 days, the soluble phosphorus content decreased in both the soil of the *P. brevicompactum*-inoculated plants and the soil of the non-inoculated plants (0.96 and 0.89 mg/kg, respectively). At 180 days, the soluble phosphorus contents in the inoculated and non-inoculated plants were 1.32 mg/kg and 1.56 mg/kg, respectively. In all samplings, the differences in available phosphorus between the plants inoculated with the *P. brevicompactum* fungus and the non-inoculated plants were not significant (*p* > 0.05) (Figure 8C).

### 2.6. Production of Mature Cherries in the Coffee Plants

At the “San Isidro” coffee plantation, 270 days after inoculation of the coffee plants with the *P. brevicompactum* strain, significant differences in bean yield were observed between the inoculated and non-inoculated plants (F = 9.95, *p* = 0.003). 

The coffee bean weight of the plants inoculated with the PSF *P. brevicompactum* was 485.13 g, which was significantly higher than that of the non-inoculated plants (277.6 g), with a difference in mean weight of 207.53 g.

At the “Los Bambus” coffee plantation, the bean weight of the coffee cherry of the inoculated plants was 53.78 g, while that of the non-inoculated plants was 50.65 g. This increase was only 3.13 g and the difference was not significant (F = 0.05, *p* = 0.81). 

At the “La Barranca” coffee plantation, a significant increase in bean yield was detected in the inoculated plants relative to the non-inoculated plants (F = 10.02, *p* = 0.0037). The mature coffee cherry bean weight of the PSF-inoculated plants was 555.73 g, while that of the non-inoculated plants was 173.86 g (Figure 9), with a difference of 381.87 g. (F = 10.024, *p* = 0.003).

## 3. Discussion

In this study, a high percentage of the total number of strains evaluated formed solubilization halos (79.5%). This result is higher than the percentage detected by Posada et al. [16], i.e., (9.73%) among strains isolated from coffee plantations in Mexico, and slightly higher than that reported by Arias et al. [17] (71.5%) in rhizosphere of coffee plants in Mexico (*Coffea arabica* var. Costa Rica). The results of this study confirm that shaded coffee plantations and tropical montane cloud forests harbor a high number of soil fungi with phosphate-solubilizing capacity. Due to the presence of different tree species native to cloud forests and the high diversity of plant species in shaded coffee plantations, these sites have a considerable accumulation of plant debris that, once decomposed, increases the humus content of the upper soil horizons. This could explain the high number of solubilizing species, given that the presence of a high population of solubilizing micromycetes has been positively related to soil organic matter content [18]. Another contributing factor could be the metabolic activity of the plant roots through exudates [19]. The large population of PSF could also be related to the presence of certain nutrients, pH, moisture content, organic matter, and some soil enzyme activities [20]. Djuuna et al. [21] found correlations between the number of PSF and the levels of soil P availability and moisture content, indicating an increase in soil P availability with a greater abundance of PSF in the soil. Some strains of filamentous fungi have been studied in research involving species of the genera *Alternaria* [22,23], *Aspergillus* [24], *Eupenicillium* [25], *Fusarium* [26], *Cladosporium* [27], *Gongronella* [28] *Helminthosporium* [29], *Mortierella* [30] *Rhizopus* [22], *Talaromyces* [28,31], and *Trichoderma* [32], prominent among which are the *Aspergillus* y *Penicillium* [32,33].

In this study, the following 21 genera presented a positive response to the solubilization of Ca_3_(PO_4_)^2^: *Anungitopsis*, *Arthrographis*, *Aspergillus*, *Beauveria*, *Gliocephalotrichum*, *Nigrospora*, *Chrysosporium*, *Cordana*, *Cylindrocarpon*, *Eladia*, *Geomyces*, *Merimbla*, *Nectria*, *Oidiodendron*, *Penicillium*, *Phialomyces*, *Phialophora*, *Pseudogliomastix*, *Sagenomella*, *Sporotrix*, and *Umbelopsis*. Several studies have indicated that the genera *Aspergillus* and *Penicillium* have a high capacity for solubilization of the phosphates of Ca, Al, and Fe [34,35,36,37,38,39].

The results obtained here corroborate the high phosphate-solubilizing capacity of *Penicillium* and *Aspergillus* species. Of the 25 *Penicillium* species evaluated, 23 formed solubilization halos, and all strains corresponding to the genera *Eupenicillium* and *Talaromyces*, which are sexual or perfect stages of *Penicillium*, were positive. Species of the *Penicillium* genus are the most studied as phosphate solubilizers, and have produced the strains that are currently used as biofertilizers., e.g., the product Fosfosol© marketed in Colombia [40,41] and mainly aimed at rice cultivation, with the active ingredient *Penicillium janthinellum*. In Canada, JumpStart^®^, produced from a strain of *Penicillium bilaiae* and tested on wheat, has been marketed since 1990. As with *Penicillium*, most of the *Aspergillus* strains tested had positive responses to solubilization. In this case, five of the six strains formed halos, with four of the strains reaching scale III. 

In this study, the range of solubilization rates was 1.5 to 3.8. These values are higher than those reported by Morales et al. [36], who reported a maximum SI of 1.3 for strains of *Penicillium albidum*, *P. thomii*, *P. restrictum*, *P. frequentans*, Gliocladium roseum, and *Penicillium* sp. Elias et al. [42] reported an SI of 2.87 for *Aspergillus* sp., Hernandez et al. [43] reported an SI of 3 for *Paecilomyces lilacinus*, and Verma and Ekka [44] reported an SI of 2.25 for *Penicillium purpureogenum*. Other studies have reported phosphate solubilization rate values of up to 5.3 SI for *Trichosporon beigelii* [45]. Romero-Fernandez et al. [46] reported values of 2.06–6.85, the latter value corresponding to the *Penicillium* strain. In a study of strains isolated from the rhizosphere of *Coffea arabica* var. Costa Rica, Arias et al. [17] presented SI values in the range of 1.13–6.5. The use of revealing tests using halos in solid culture media and SI data is a tool with which to detect the phosphate-solubilizing capacity of strains rapidly and easily; however, the formation of halos in solid media should not be considered to be the only test for evaluating solubilization capacities. Quantitative tests are also necessary since Arias et al. [17] did not find a significant relationship between the SI in a solid medium and the solubilized phosphorus content in a liquid medium. 

Although the *A. niger* strain showed a lower SI than *Sagenomella diversispora*, *P. waksamanii,* and *P. brevicompactum*, this strain was chosen for quantitative evaluation because, on the one hand, in addition to having rapid growth, the solubilization halo was detected on the first day of inoculation, so it was considered to have high solubilizing potential. On the other hand, the *S. diversispora* strain was not chosen since it is a slow-growing strain.

In the quantitative evaluation, the absence of soluble P in the controls clearly demonstrated that all three strains tested are phosphate solubilizers and that the manipulation of the cultures was appropriate. The *A. niger* strains solubilized a high amount of insoluble phosphorus at 98.22 mg/L, followed by *P. waksmanii* at 95.77 mg/L, and *P. brevicompactum* at 74.63 mg/L. The values presented here exceed those highlighted in other studies [17,36,37,43,47,48,49,50]; however, they were lower than the values presented for the *Aspergillus* sp. strain (167.7 mg/L) [37].

Several authors have argued that fungal species use the production of organic acids as a solubilization mechanism, reflected in a reduced pH [51,52,53]. The data presented here showed that Ca_3_(PO_4_)^2^ could be solubilized by lowering the pH to 1.9. PSF have been shown to play a crucial role in phosphorus solubilization due to the production of organic acids. *Aspergillus niger* and some *Penicillium* sp. have been studied concerning the solubilization of rock phosphate and phosphate mobilization [35,54]. Acidification of the growth medium through the production of organic acids can be explained by the utilization of glucose as a carbon source during growth. Similar results have been extrapolated from previous studies [55]. Organic acids secreted by fungi dissolve mineral phosphate because of the anion exchange of PO_4_^3−^ for the acid anion [56].

Seven different organic acids (salicylic, ascorbic, citric, formic, lactic, oxalic, and malic acid) were detected during in vitro phosphate solubilization. Solubilization by acidification depends on the nature and strength of the acids produced by the fungus since there are those, such as tri- or dicarboxylic acids, that are more efficient solubilizers than mono-basic and aromatic acids [43,57,58]. Salicylic acid was the most frequently produced organic acid by all the PSF. Its concentration was 1823.8 mg/l in the case of *Aspergillus japonicus*. Salicylic acid was the most frequently produced organic acid among all the PSF isolates, having important roles in defense mechanisms against biotic and abiotic stresses [53].

Some studies have reported that enzymes, such as acid phosphatase and phytases, are involved in P solubilization [57,59,60,61,62]. These enzymes cleave the ester bond of the insoluble forms of P, leaving orthophosphate ions that can be assimilated by plants [63]. Previous reports indicate that the genera *Aspergillus* and *Penicillium* both actively participate in the mineralization of soil organic P through the production of phosphatase enzymes [64].

The results presented here evidence the production of acid phosphatase by the solubilizing fungi of the two strains of the genus *Penicillium* and the strain of *A. niger*. The highest production of acid phosphatases was detected in the *P. brevicompactum* strain. These results support those obtained by Gomez [65], who found that some *Penicillium* species may have a higher capacity to produce acid phosphatases than *Aspergillus* species. Similar reports were obtained by Valenzuela et al. [66] who evaluated 180 fungal strains and found that *Penicillium* strains showed a higher activity of this enzyme than that produced by *Aspergillus* strains.

This study shows that the *A. niger* strain was superior to the other strains in terms of its ability to solubilize tricalcium phosphate. Other studies report that *A. niger* produces oxalic acid [67] which could explain its higher P solubilization capacity compared to the other microorganisms. Solubilization in the *P. brevicompactum* strain may be mostly achieved through the production of phosphatase enzymes since the results show that this strain presented the highest content of these. 

Microorganisms harbored in the rhizosphere of plants play several key roles in improving plant productivity, nutrient cycling, and soil fertility. Overall, in this field study conducted on three coffee plantations in Jilotepec, Veracruz, inoculation with the phosphate-solubilizing fungus strain *P. brevicompactum* increased productivity in the coffee plants. The phosphate-solubilizing potential of the *P. brevicompactum* strain had already been assessed on coffee (*Coffea arabica* var. Garnica) plants by Perea et al. [68]; however, that study was conducted under controlled conditions. It is important to note that due to this background and for comparative purposes, this strain was chosen for field inoculation. In this study, the inoculation of PSF (both alone and together with mycorrhizae) favored the development and availability of phosphorus in the soil, as well as the foliar phosphorus content. However, given that the study was carried out under controlled conditions and with 8-month-old seedlings, it was not possible to determine the effect of PSF on coffee bean production.

Concerning the soil soluble phosphorus content, this parameter increased after two months of inoculation with *P. brevicompactum*. In the second sampling (fourth month of inoculation), however, it had significantly reduced, although the values remained higher in the plants inoculated with *P.* brevicompactum in two plantations. In “La Barranca, for all samplings, the inoculated and non-inoculated plants showed no difference in the soluble phosphorous content; it is likely that the acidic soil condition influenced this result.

In the first sampling, the foliar phosphorus content increased, but by the second sampling, it had reduced significantly. In the third sampling, the values increased, and they were significantly greater than the initial content. In all plantations, the foliar phosphorus values of the non-inoculated plants were slightly higher; however, the difference was not significant.

The subsequent decrease in both soil and leaf phosphorus contents may have been because the solubilized phosphorus was being used by the plant in coffee cherry production. In a study in North Bengal in India, Chakraborty et al. [69] evaluated the ability of some PSF to promote plant growth in soybean seedlings and found a reduction in phosphorus in the soil. In that study, the concentration of phosphorus was significantly increased in the roots. It is, therefore, recommended that further research be conducted to monitor the phosphorus in plants from the roots to the leaves.

Several studies have documented the successful use of bioinoculants, which have been used as an alternative to traditional fertilizers. A study in Australia [39] used bioinoculants based on *Penicillium bilaiae* and *Penicillium radicum,* which showed beneficial effects on wheat crops. In Canada, a biofertilizer based on the fungus *Penicillium bilaiae* was registered in 1990 for use on wheat and tested initially on a few hectares with good results, and in 2002, approximately one million hectares sown with principal crops in Canada used this biofertilizer [70].

In Colombia, a biofertilizer with phosphate-solubilizing microorganisms, the active ingredient of which is *Penicillium janthinellum*, has also been marketed. This product has been used in rice cultivation, producing high increases in the production of this cereal [41]. The results obtained in this study are promising for the development of a bioinoculant of the fungal strain *P. brevicompactum*; however, further research is required to establish appropriate doses and re-inoculation schedules. The vigorous microbial activities in the soil optimize nutrient cycling and maximize the efficiency of their use in agronomy [71]. A recent extensive review [72] of greenhouse and field trials showed a marked improvement in the growth responses of various crops to inoculation with phosphate-solubilizing microorganisms. 

Most studies have been conducted with bacteria; however, although they are the main decomposers of organic matter and dictate soil carbon and other elements, the role of fungi is the subject of relatively few studies. Many fungi can solubilize insoluble phosphates or facilitate P acquisition by plants; and therefore, they form an important part of commercial microbial products, with *Aspergillus, Penicillium*, and *Trichoderma* being the most efficient [73].

The use of bioinoculants, together with the rationalized use of phosphate fertilizers, is important, and it is necessary to adjust fertilization levels (especially of P and N) to reduce negative impacts on the environment. Chemical-based agriculture has had a negative impact on beneficial microbial communities, significantly reducing microbial biodiversity; therefore, there is a need to adopt ecological farming practices for sustainable agriculture. Indigenous or native microorganisms are considered to be an important tool with which to overcome problems associated with the overuse of chemical fertilizers and pesticides.

Although research has been conducted on plant growth promotion using phosphate-solubilizing bacteria and fungi in vitro, particularly in coffee (*Coffea arabica* L.), there are few reports regarding their impact on growth under field conditions. This study is the first, worldwide, to evaluate the potential of this group of microorganisms in coffee bean production.

## 4. Materials and Methods

The strains originated from coffee plantation systems and tropical montane cloud forest soil, in 2004–2005, in the central region of the state of Veracruz, Mexico [74]. These strains were preserved in distilled water and deposited in the Collection of the Micromycetes Laboratory of the Ecology Institute, Xalapa, Veracruz, México. The identification method was traditional taxonomy.

### 4.1. Determination of Solubilization Index

The strains were reactivated in potato dextrose agar (PDA) medium and subjected to a screening test for their phosphate solubilization potential. Sundara medium was used, which contained (NH_4_)_2_SO_4_ (0.5g), KCl (0.2 g), MgSO_4_·7H_2_O (0.3 g), MnSO_4_·H_2_O (0.004 g), FeSO_4_·7H_2_O (0.002 g), NaCl (0.2 g), D-glucosa (10 g), yeast extract (0.5 g), chloramphenicol (0.1 g), agar (20 g), gum arabic (0.5 g), Ca_3_(PO_4_)_2_ (0.5 g), and water (1000 mL). After 72 h of incubation at 25 °C, the formation of a halo around the fungal growth on the medium indicated phosphate solubilization. 

To categorize the solubilization of the strains that formed halos, three levels were established according to the thickness of the halo. The levels were as follows: level I colonies with halos of 1–4 mm, level II colonies with halos of 5–8 mm, and level III colonies with halos of 9–12 mm.

The solubilization index (SI) was measured only in strains categorized as level III; it was calculated as the ratio of the total diameter (colony  +  halo zone) to the colony diameter [75]. For the selection of strains with a higher SI, the scale established by Silva Filho and Vidor [15] was used. In this scale, the solubilizing capacity is considered to be low when the SI is lower than 2, medium if it is higher than 2 and lower than 3, and high if it is higher than 3.

### 4.2. Phosphate Solubilization Efficiency of Fungal Isolates

For this test, three strains were selected from the high solubilization range. These strains were inoculated into Sundara liquid medium [76] supplemented with 0.5 g^−1^ tricalcium phosphate as a source of insoluble phosphorus. The strains were inoculated with 5 mm diameter mycelium discs, with 8 days of growth on solid PDA, in triplicate. Negative controls were established using discs of medium with no inoculum. The cultures were incubated in darkness at 25 °C for 21 days. In each sampling, the pH of each sample was recorded, and the biomass achieved for each fungus was determined. To obtain the fungal extracts, the samples were filtered with Whatman^®^ 42 filter paper (GE Healthcare, General Electric Company, Puebla, Mexico) and the soluble phosphorus was quantified. The pH was measured with a potentiometer. Soluble phosphorus was measured with the ascorbic acid method by Clesceri et al. [77], while absorbance was measured with a spectrophotometer at 880 nm. The data were compared with a standard curve for phosphorus and expressed in mg/mL.

### 4.3. Phosphatase Activity Assay 

Acid phosphatase activity was monitored according to Tabatabai and Bremner [78]. In total, 900 μL of enzyme extract, 90 μL of 1M acetate buffer (pH 5), and 10 μL of 15 mM p-nitrophenyl phosphate were mixed and incubated for 1 h at 37 °C. The results were expressed in UAE/mg protein.

### 4.4. Characterization of Isoenzymatic Patterns

The electrophoretic profiles were analyzed using polyacrylamide gels with the following modifications: 4% polyacrylamide stacking gel and 10% separating gel. Electrophoresis was run in a vertical electrophoresis chamber with 19 mM tris-glycine buffer of pH 8.3 at 25 mA and 4 °C for 2 h. Each well contained 20 μg of protein. The acid phosphatase activity was measured in 500 mM citrate buffer at pH 5.5 with β naphthyl sodium phosphate and fast black K salt as a substrate. The gel was incubated at 37 °C in darkness with agitation (125 rpm) for 4 h.

### 4.5. Inoculation of a PSF Strain in Coffee Plants under Field Conditions

The coffee plantations where the solubilizing fungi were inoculated are located in Jilotepec, Veracruz. For this study, three shaded coffee plantations were selected, with coffee plants of *Coffea arabica* var. Costa Rica (Figure 10). 

The geographical characteristics, elevation, precipitation, temperature, and soil type of the sites are listed in Table 1. 

Within each of the three plantations of approximately 2 ha, plots of 500 m^2^ were delimited (plot 1/plot 2). In each plot, 25 coffee plants (*Coffea arabica* L.) of the Costa Rica variety were marked and sampled in “cinco de oros” [79]. In each plantation, Plot 1 was used for inoculation of the PFS strain, while Plot 2 represented the control treatment (non-inoculated) (Figure 11).

Strain RA103 (*Penicillium brevicompactum*) was selected for its capacity to solubilize tricalcium phosphate in vitro, as well as its fast growth. It was propagated on potato dextrose agar medium and incubated for 1 week at 25 °C. Spores were then scraped from the medium and suspended in a solution containing 10 mL of Inex A to create a solution of 1 × 10^10^ spores/mL. 

In Plot 1 of each plantation, 10 mL of fungal inoculant (*Penicillium brevicompactum*) was applied to each of the 25 plants using a syringe. These applications were carried out at the base of each plant near the stem and on the roots. 

### 4.6. Variable Measurement

At the beginning of the experiment (July 2021), and every 2 months (3 samplings) thereafter, soil samples (500 g) were taken to measure the soluble phosphorus content and leaf samples (10 leaves per plant) were collected to measure foliar phosphorus. In addition, during March and April 2022, coffee bean production was measured.

The collected soil was dried at ambient temperature and passed through sieves (DAIGGER ATM 25SS8F, 800-621-7193, Daigger Scientific, Mexico City) to remove stones. The collected coffee leaves were dried in an oven (BINDER 09-08078, Binder Inc Mx, Mexico City, Mexico) at 60 °C for approximately 8 h and ground in a mill before the measurement of leaf phosphorus. 

Leaf phosphorus was measured following the technique of McKean [80]. For this, 0.25 g of dried leaves was weighed and placed in porcelain crucibles inside a muffle furnace at 500 °C for 2 h. The resulting ashes were dissolved in 25 mL of 0.3 M HCl in test tubes. The extracts were filtered through Whatman filter paper (42). After filtration, 18 mL of the combined reagent was added to 2 mL of each sample and allowed to stand for 20 min. The samples were then measured with a spectrophotometer at 660 nm. 

Soluble phosphorus was quantified using the technique of Bray and Kurtz [81]. For this, 1.2 g of soil was weighed, and 10 mL of the extractant solution (0.03 N ammonium fluoride, hydrochloric acid, and distilled water) was added. The extracts were transferred to Whatman filter paper. Subsequently, a 1 mL aliquot of extracted sample was taken, 5 mL of combined reagent (ascorbic acid in stock solution) was added to the sample, and the sample was made up to 50 mL with distilled water. The samples were measured with a spectrophotometer (JENWAY, model 6305, Bibby Scientific, Fisher Scientific UK Ltd., England) at 882 nm, using distilled water as a blank. To measure the effect of PSF inoculation on coffee bean production, ripe fruits were cut in both plots 270 days after inoculation, and their weight (gr) was recorded using an analytical balance.

### 4.7. Physicochemical Analysis of the Soil

Physicochemical analyses of the rhizospheric soil of the coffee plantations were conducted according to NOM 021-RECNAT-2000 (Table 2). Organic matter (OM) and organic carbon (OC) were quantified following the modified Walkley–Black method. The pH was measured using the electrometric method. Cation exchange capacity (CEC) was measured using 1N ammonium acetate (pH 7.0). Total nitrogen was measured using the micro-Kjeldahl method. Retained phosphorus was measured using the Blakemore method [82]. These analyses were conducted at the Soil, Plant, and Water Analysis Laboratory of the Instituto de Ecologia, A.C.

### 4.8. Statistical Analysis of Data

The data were analyzed using one-way ANOVA and Fisher’s LSD means tests to determine significant differences between groups at *p* < 0.05. All statistical analyses were performed using Statistica version 7.0 [83].

## Figures and Tables

**Figure 1 plants-12-03395-f001:**
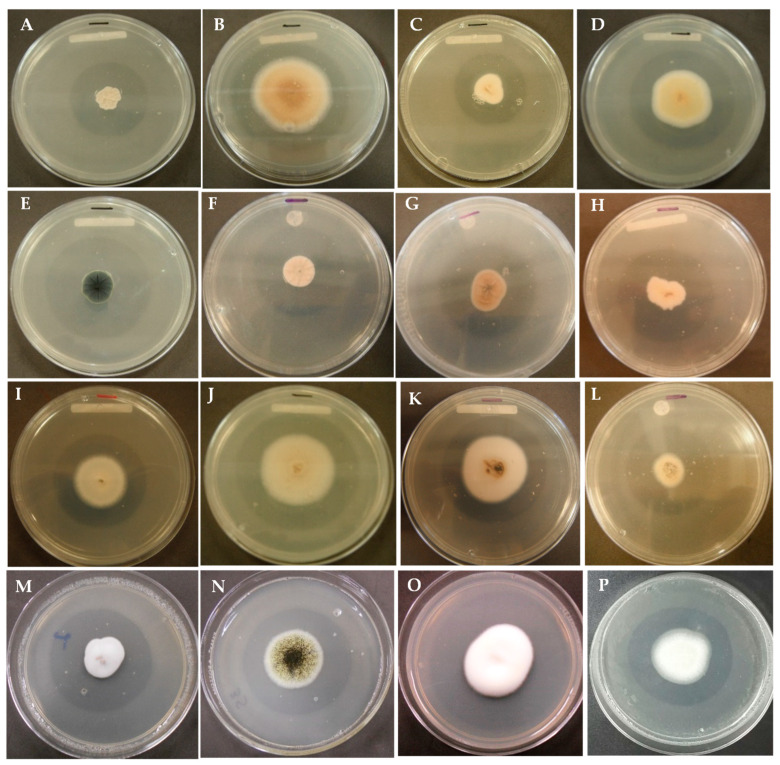
Fungi showing phosphate solubilization leading to formation of clear zone in Sundara’s medium: (**A**) *Aspergillus candidus*; (**B**) *Aspergillus sclerotiorum*; (**C**) *Acremonium roseolum*; (**D**) *Aspergillus* sp. 1Y; (**E**) *Cladosporium cladosporioides*; (**F**) *Eupenicillium euglaucum*; (**G**) *Eupenicillium ludwigii*; (**H**) *Fusarium* sp. 25; (**I**) *Penicillium arenicola*; (**J**) *Penicillium glabrum*; (**K**) *Penicillium waksmanii*; (**L**) *Sagenomella diversispora*; (**M**) *Paecilomyces marquandii*; (**N**) *Aspergillus niger*; (**O**) *Beauveria bassiana*; (**P**) *Penicillium brevicompactum*.

**Figure 2 plants-12-03395-f002:**
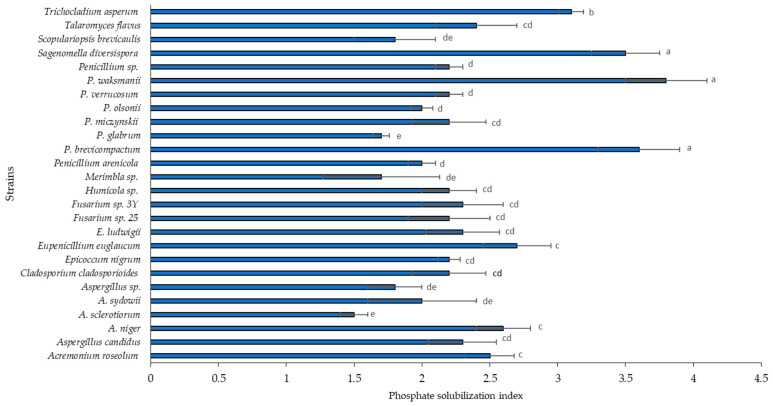
The phosphate solubilization index of 26 fungal strains ubicated in level III halo thickness (9–12 mm in thickness). Significantly different means (*p* < 0.05) from the one-way ANOVA followed by LSD. Identical letters in the rows indicate no significant difference.

**Figure 3 plants-12-03395-f003:**
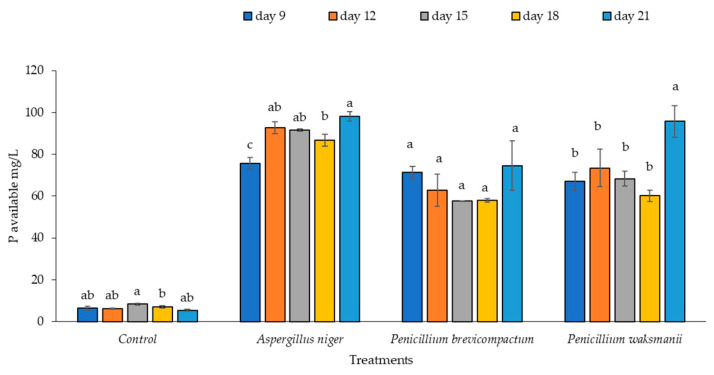
Soluble phosphorus from extracts of three PSF strains (*A. niger*, *P. brevicompactum*, and *P. waksamnii*) and the control, over 21 days of incubation (measured at 9, 12, 15, 18 and 21 days). Data are the average of three replicates ± standard error. Significantly different means (*p* < 0.05) from the one-way ANOVA followed by LSD. Identical letters in columns indicate no significant difference.

**Figure 4 plants-12-03395-f004:**
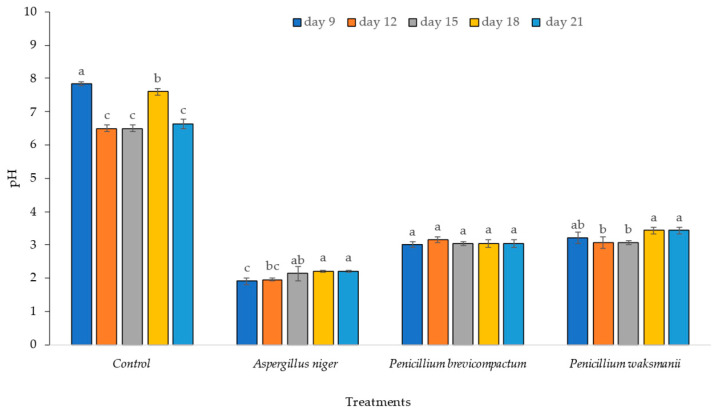
pH from extracts of three PSF strains (*A. niger*, *P. brevicompactum*, *P. waksamnii*) and the control, over 21 days of incubation (measured at 9, 12, 15, 18 and 21 days). Data are the average of three replicates ± standard error. Significantly different means (*p* < 0.05) from the one-way ANOVA followed by LSD. Identical letters in the columns indicate no significant difference.

**Figure 5 plants-12-03395-f005:**
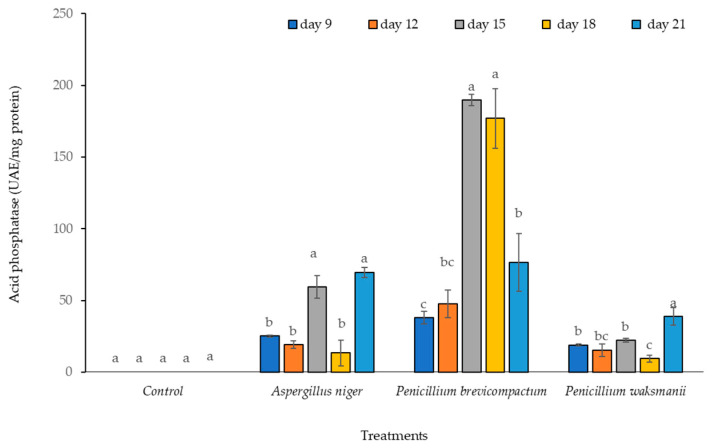
Acid phosphatase from extracts of three PSF strains (*A. niger*, *P. brevicompactum*, and *P. waksamnii*) and the control, over 21 days of incubation (measured at 9, 12, 15, 18 and 21 days). Data are the average of three replicates ± standard error. Significantly different means (*p* < 0.05) from the one-way ANOVA followed by LSD. Identical letters in the columns indicate no significant difference.

**Figure 6 plants-12-03395-f006:**
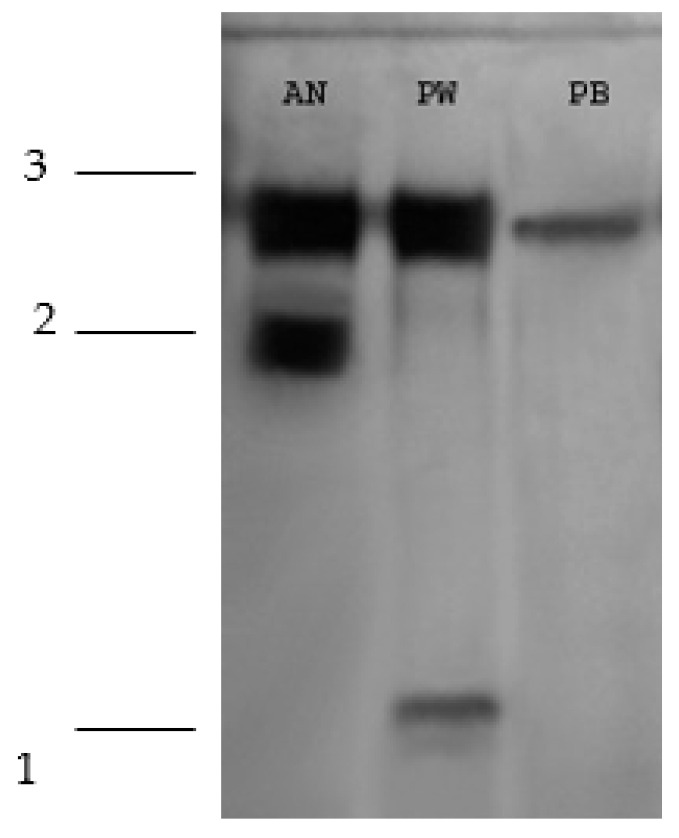
Isoenzymatic patterns of in vitro acid phosphatases of the strains (*A. niger* AN, *P. brevicompactum* PB, and *P. waksamnii* PW).

**Figure 7 plants-12-03395-f007:**
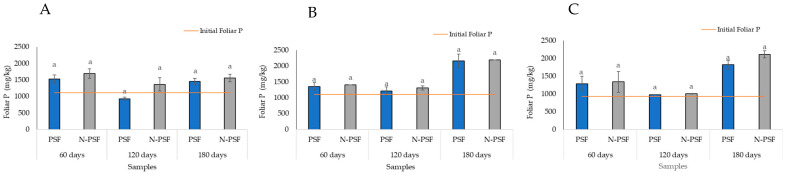
Foliar phosphorus content of coffee (*Coffea arabica* var. Costa Rica) plants from the “San Isidro (**A**), “Los Bambus” (**B**), and “La Barranca” (**C**) coffee plantations, in Jilotepec. PSF, Plants inoculated with the phosphorus solubilizing fungus (*P. brevicompactum*); N-PSF, uninoculated plants at 60, 120 and 180 days. Data are the average of fifteen replicates ± standard error. Significantly different means (*p* < 0.05) from the one-way ANOVA followed by LSD. Identical letters in the columns indicate no significant difference.

**Figure 8 plants-12-03395-f008:**
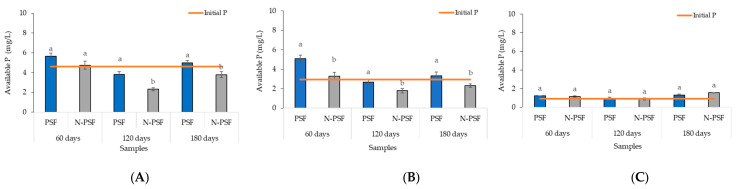
Soil soluble phosphorus content of coffee (*Coffea arabica* var. Costa Rica) plants from the “San Isidro (**A**), “Los Bambus” (**B**), and “La Barranca” (**C**) coffee plantations, in Jilotepec. PSF, Plants inoculated with the phosphorus solubilizing fungus (*P. brevicompactum*); N-PSF, uninoculated plants at 60, 120 and 180 days. Data are the average of fifteen replicates ± standard error. Significantly different means (*p* < 0.05) from the one-way ANOVA followed by LSD. Identical letters in the columns indicate no significant difference.

**Figure 9 plants-12-03395-f009:**
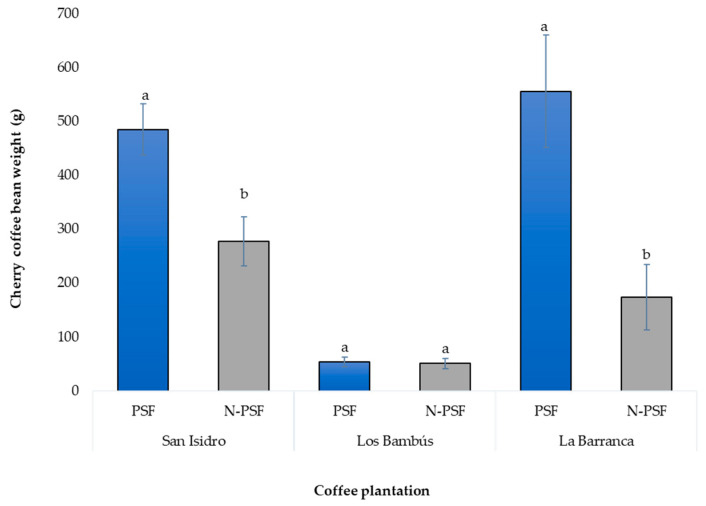
Production of cherry coffee beans from plants of *Coffea arabica* var. Costa Rica of three coffee plantations. PSF, Plants inoculated with the phosphorus-solubilizing fungus *P. brevicompactum*; N-PSF, plants without the phosphorus-solubilizing fungus. Data are the average of fifteen replicates ± standard error. Significantly different means (*p* < 0.05) from the one-way ANOVA followed by LSD. Identical letters in the columns indicate no significant difference.

**Figure 10 plants-12-03395-f010:**
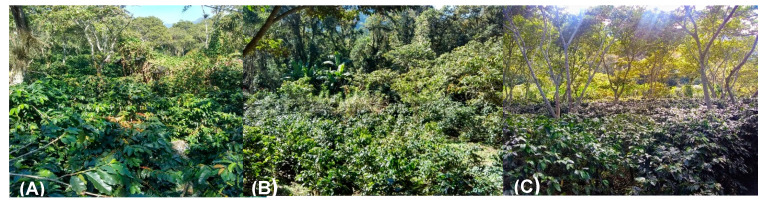
Selected coffee plantations of central Veracruz state, Mexico: (**A**) “San Isidro”; (**B**) “Los Bambus”; (**C**) “La Barranca”.

**Figure 11 plants-12-03395-f011:**
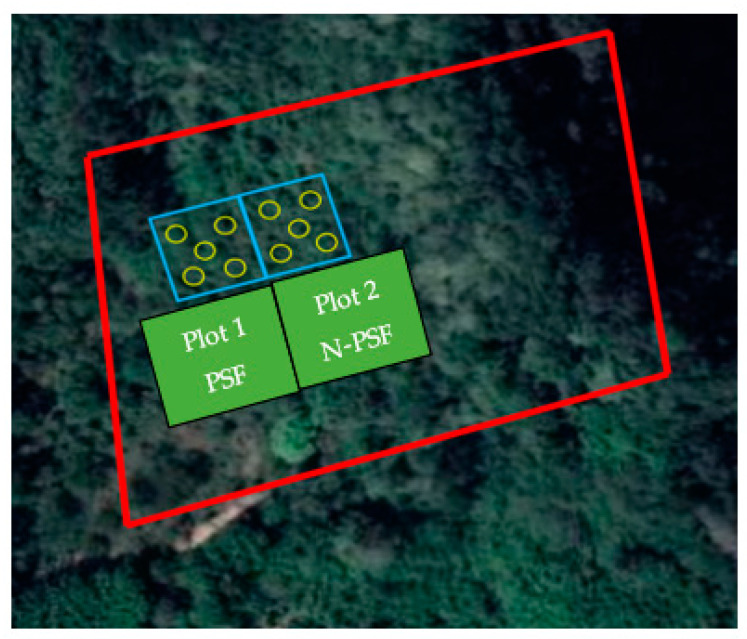
Establishment design of the field experiment. Plots of 500 m^2^ (Plot 1/Plot 2). Plot 1 was used for inoculation of the PFS strain, while Plot 2 represented the control treatment (N-PSF).

**Table 1 plants-12-03395-t001:** Geographic location, elevation, and characteristics of the study sites.

Sites	Annual Mean Precipitation	Latitude	Longitude	Elevation (MSL)	Mean Temperature	Management Type	Soil Type
**San Isidro**	241	19°36′42.74″	96°56′16.01″	1370	22	Traditional polyculture	Andosol
**Los Bambus**	275.9	19°36′38.07″	96°55′40.57″	1414	25	Traditional polyculture	Andosol
**La Barranca**	298.2	19°36′12.15″	96°54′44.91″	1484	25	Traditional polyculture	Andosol

**Table 2 plants-12-03395-t002:** Physicochemical characteristics of the coffee plantations evaluated.

	Coffee Plantations		
	“San Isidro”	“Los Bambus”	“La Barranca”
pH	6.11	6.69	5.43
Retained P	87.35	89.8	81.63
Organic material	12.46	3.93	4.72
Organic carbon	7.23	2.28	2.74
Cation exchange capacity (CEC)	27.09	20.88	21.51
Field capacity (FC)	31.72	22.69	21.62
Bulk density	0.893	1.016	0.994
Clay	29.8	45.8	49.8
Silt	30.56	22.56	28.56
Sand	39.64	31.64	21.64
Texture	Clay loam	Clay	Clay
Fe	69.38	111.63	163.03
C	8.5	2.9	3.5
N	0.72	0.27	0.27

## Data Availability

Not applicable.

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
