# Peer review of "Selection and Characterization of Phosphate-Solubilizing Fungi and Their Effects on Coffee Plantations"

_plants, 2023, doi:10.3390/plants12193395_

Round 1

Reviewer 1 Report

1)     Your manuscript ¨Selection and characterization of phosphate solubilizing fungi and their effects on coffee plantations¨ is an interesting study but needs major revisions before being accepted for publication.

2)     Generally, English language needs to be improved throughout your manuscript.

a.      e. g. The use of phosphate-solubilizing fungi in coffee cultivation is an alternative with which  to reduce the use of fertilizers?. There are repetitive mistakes throughout the manuscript.

b.     Use the term ¨phosphate solubilizing¨ in consistent style. I.e. phosphate-solubilizing.

3. Results

1.     The data need statistical analysis e.g. Table 1.

2.     Figures: The axis labels, tick marks etc are invisible. Their resolution should be increased. Add the statistical test used in footnotes.

3.     Why was root colonization of strains not assessed? It is an important parameter that should be determined in this study.

4. Materials and methods

1.     The source of strains should be added.

 English language needs to be improved throughout your manuscript.

Author Response

Consulte el archivo adjunto.

Reviewer 2 Report

Title: Selection and characterization of phosphate solubilizing fungi and their effects on coffee plantations

Isolation, characterization and application of microbes based on its functionality promotes organic agriculture. Organic agriculture is the future for sustainable agriculture reduction. In this regards, many studies have isolated and characterized for their functional aspect in terms of use in agriculture production. Most of the studies have focused on bacteria as bio inoculants. Few studies were also characterized fungi as bio inoculant. Chemical-based agriculture has had a negative impact on beneficial microbial communities, significantly reducing microbial biodiversity. Hence, therefore a need to adopt ecological farming practices for sustainable agriculture by exploiting beneficial native inoculants. As introduced beneficial microbes as to compete with native isolates. Most of the cases, the introduced organisms fail to establish the population by competing native isolates. Indigenous or native microorganisms are considered an important tool to overcome problems associated with the overuse of chemical fertilizers and pesticides.

Although many research has been conducted using phosphate-solubilizing bacteria and fungi in vitro on plant growth promotion in many crops, particularly in coffee (Coffea arabica L), there are few reports regarding their impact on growth under field conditions. This study is the one among them evaluated the potential of native beneficial fungi coffee bean production under field conditions.

Overall, manuscript content is interest to the readers and potential commercial application of microbes identified in this study. However, I have some suggestions to improve the presentation of this manuscript. 

Abstract:

Line 21, of the 151 strains analyzed, Aspergillus niger, it is not coming under scale III based on classification mentioned in materials and methods section.

Introduction:

This section gives a clear background on functional role of microbes for sustainable agriculture production, particularly coffee production as well as ways to improve the productivity by maintaining soil fertility.

Line 40, 750 MSL.

Results:

Table 1.

Line 146-154, Sagenomella diversispora and Trichocladium asperum not included in the classification.

Line 155, Aspergillus niger, is it not under scale III as per the values given in the table 1.

Lines 176-200, Figure 2: Presentation is not clear and difficult to follow. Values given in the text is not matching with values in the graph. Please verify. Labelling is not done properly. Difficult to follow which is AN, PB, PW and C. Why control is given separately. It should be with AN, PB and PW for easy comparison and understanding.

Line 211-223, Y axis unit is not correct

Line 253-282, Why, 1. Non-inoculated Control samples have high soluble phosphorous in the leaf compared to inoculated; 2. Always 120 days inoculation and non-inoculation samples showed lower available phosphorous compared to 60 and 120 days after inoculation; and 3. Figure 5. Labelling should be proper, indicate what is A, B and C.

Line 326, Figure 20??

Line 329-338, Figure 6. Labelling should be proper, indicate what is A, B and C.

Line 319-329, Why, in highly acidic soil conditions (La Barranca), both inoculated and non-inoculated showed no difference in soluble phosphorous content.

Discussion:

Avoid typographical errors in this section.

Line 418, verify the scale III

Line 419, In this study, the range of solubilization rates (RSR) was 1.14 to 3.88.

Line 425, reported 2.25

Line 426, reported of 2.06-6.85

Line 443, solubilized by lowering the pH of 1.9

Line 446, rock phosphate and phosphate mobilization

Line 456, Not listed in the table 1.

Line 451-478, may be made as one paragraph

Line 478, highest content of phosphatase enzyme

Line 531, therefore a need to adopt ecological

Materials and methods

Line 543, Sundara medium [76] containing

Line 571, correct the units

Line 577, check the unit ºC

Whether the Phosphorous content measured before the inoculation of the isolates?

Whether the PSF population is counted after the inoculation?

Line 642, Blakemore method, include reference

Line 646-647, Table 4, font is not in required format

Author Response

Consulte el archivo adjunto.

Reviewer 3 Report

1. add more detail of the strains in “Materials and Methods”, such as collecting location, time, address of strain preservation and method of identification.

2.please explain choose the stain Penicillium brevicompactum for coffee plantation.

3. in table 1, Aspergillus niger only 3 sample?

4. add symbol of each picture in Figure 2, 5, 6.

5. in line 419,  the range of solubilization rates (RSR) maybe 1.2 to 3.8

Round 2

Reviewer 1 Report

Revised version is improved and acceptable.